# Molecular Targets of Genistein and Its Related Flavonoids to Exert Anticancer Effects

**DOI:** 10.3390/ijms20102420

**Published:** 2019-05-16

**Authors:** Hee-Sung Chae, Rong Xu, Jae-Yeon Won, Young-Won Chin, Hyungshin Yim

**Affiliations:** 1College of Pharmacy, Dongguk University-Seoul, Goyang, Gyeonggi-do 10326, Korea; chaeheesung83@gmail.com (H.-S.C.); f2744@dongguk.edu (Y.-W.C.); 2Department of Pharmacy, College of Pharmacy, Institute of Pharmaceutical Science and Technology, Hanyang University, Ansan, Gyeonggi-do 15588, Korea; heoyong310@gmail.com (R.X.); waeyoen@gmail.com (J.-Y.W.)

**Keywords:** flavonoid, genistein, PLK1, anticancer

## Abstract

Increased health awareness among the public has highlighted the health benefits of dietary supplements including flavonoids. As flavonoids target several critical factors to exert a variety of biological effects, studies to identify their target-specific effects have been conducted. Herein, we discuss the basic structures of flavonoids and their anticancer activities in relation to the specific biological targets acted upon by these flavonoids. Flavonoids target several signaling pathways involved in apoptosis, cell cycle arrest, mitogen-activated protein kinase (MAPK), phosphoinositide 3-kinase (PI3K)/AKT kinase, and metastasis. Polo-like kinase 1 (PLK1) has been recognized as a valuable target in cancer treatment due to the prognostic implication of PLK1 in cancer patients and its clinical relevance between the overexpression of PLK1 and the reduced survival rates of several carcinoma patients. Recent studies suggest that several flavonoids, including genistein directly inhibit PLK1 inhibitory activity. Later, we focus on the anticancer effects of genistein through inhibition of PLK1.

## 1. Introduction

Flavonoids, a class of natural polyphenolic compounds, are commonly found in fruits, soybeans, and vegetables; medicinal plants containing flavonoids are also used in herbal medicine [1]. With an increase in public health awareness, dietary supplements including flavonoids have gained popularity due to their beneficial biological activities. Flavonoids have been intensively studied to understand the specific proteins targeted by them to exert anticancer [2,3,4,5,6], anti-inflammatory [7], anti-diabetic [8,9], anti-oxidant [10], cardio-protective [11], neuroprotective, and cognitive enhancing activities [12]. Anticancer activities induced by flavonoids are a result of their effects on several signaling pathways involved in apoptosis, cell cycle arrest, MAPK kinase, PI3K/AKT kinase, and metastasis. In the first part of this review, we focus on the basic structure and biological activities of flavonoids having anticancer effects.

As one of the promising molecular targets for anticancer drug development, Polo-like kinase 1 (PLK1) has recently gained importance [13,14,15,16,17]. PLK1 is a serine/threonine mitotic kinase having a polo-box domain for mitotic regulation such as mitotic entry, centrosome maturation, spindle pole formation, and chromosomal segregation in cells [17,18]. PLK1 is overexpressed in several carcinomas, and its expression correlates with tumor proliferation and malignancy [17]. Its expression is inversely correlated with survival rates of breast, colon, lung, and prostate cancer patients [16,17]. Due to the clinical relevance between PLK1 expression and cancer malignancy, PLK1 has been investigated as a target for anticancer drug development [16,17]. Until now, several PLK1 inhibitors including BI 2536, volasertib, GSK461364, and rigosertib have been identified through in vitro and in vivo experiments [16,19,20]. Volasertib, developed by Boehringer Ingelheim Pharma (Germany), is now being tested in a Phase III clinical trial for leukemia that initiated in 2017 (NCT01721876). Rigosertib, a multitarget inhibitor of PLK1 and PI3K [16,19,21], developed by Oncova Therapeutics (USA), is being tested in a Phase III clinical study for chronic myelomonocytic leukemia (NCT01928537).

Flavonoids from natural products have PLK1 inhibitory effects. Genistein, baicalein, 7-0-methylwogonin, dihydrobaicalein, and viscidulin II have PLK1 inhibitory activity [22,23]. Among them, genistein is more selective and effective to PLK1 compared with other flavonoids. In the latter part of this review article, we discuss the PLK1 inhibitory activities of flavonoids especially those of genistein. In addition, the anticancer effects of genistein through its effects on diverse targets are reviewed, since genistein has been broadly studied for chemoprevention and chemotherapy.

## 2. Flavonoids

### 2.1. Structures of Flavonoids

Flavonoids commonly contain a C6-C3-C6 backbone, which may be 2-phenylchromane (flavonoid), 3-phenylchromane (isoflavonoid), and 4-phenylchromane (neoflavonoid) [24]. Among them, flavonoid and isoflavonoids have been widely investigated [5,24] Flavonoids are further divided into sub-classes depending on the presence or absence of double bonds, carbonyl groups, and position of the hydroxyl group. Flavanone contains a carbonyl group, whereas flavan does not. If the flavanone has a double bond in the C-ring, it becomes a flavone. Based on the position of the hydroxyl group in the A, B, and C-rings, these structures can be further sub-classified (Figure 1). The flavone structures are divided into two groups depending on the presence of a hydroxyl group in the C-ring: flavone (no hydroxyl group in the C-ring) and flavonol (a hydroxyl group in the C-ring). Common flavones are chrysin (5,7-dihydroxyflavone), apigenin (5,7,4′-trihydroxyflavone), luteolin (5,7,3′,4′-tetrahydroxyflavone), and tricetin (5,7,3′,4′,5′-pentahydroxyflavone). Flavonols include galangin (3,5,7-trihydroxyflavone), kaempferol (3,5,7,4′-tetrahydroxyflavone), quercetin (3,5,7,3′,4′-pentahydroxyflavone), and myricetin (5,7,3′,4′,5′-hexahydroxyflavone). Flavanones are also subdivided into two classes: flavanone (e.g., naringenin, eriodictyol) and flavanonol (e.g., taxifolin). As flavan-type flavonoids, catechin and its analogs are mostly well-known and found in *Camellia* species.

Isoflavonoids such as genistein and daidzein are distributed in *Leguminosae* species, including soybeans. They contain differently positioned hydroxyl groups, methylation, prenylation, and glycosylation over the basic flavonoid structure. They are found in *Scutellaria baicalensis* (e.g., baicalein, oroxylin A, scutellarein, wogonin, 7-*O*-methylwogonin, viscidulin II and dihydrobaicalein), *Morus alba* (e.g., morin), and *Citrus* species (e.g., nobiletin), as shown in Figure 1.

### 2.2. Anticancer Effects of Flavonoids

Anticancer effects of flavonoids have been well documented elsewhere [25,26,27], and herein, we present recent findings on the modulation of oncogenic pathways by representative flavonoids. These flavonoids have been reported to suppress carcinogenesis in various cancer cell models in vitro and in vivo [26]. By targeting multiple cancer pathways such as cell metabolism, apoptosis, adhesion, migration, angiogenesis, and immune response [28], flavonoids, including flavones (apigenin, luteolin, nobiletin, baicalein), flavanols (quercetin, kaempferol, myricetin, fisetin, morin), flavanones (hesperidin, hesperetin, naringin, naringenin), and isoflavones (genistein, daidzein) have been found to exhibit potent anti-proliferation effects on cancer cells [29,30]. The flavonoids listed in Table 1 were selected in this review because they are consumed abundantly through the diet and have shown promising anticancer activities. In addition, the common signaling pathways triggered by these flavonoids were displayed in Table 2.

#### 2.2.1. Flavones with Anti-Cancer Effects

##### 2.2.1.1. Apigenin

Apigenin, found in several types of berries and vegetables, regulates pathways involved in the progression of gastric cancer [31], colon cancer [32], and hepatocellular carcinoma [33]. The maximum solubility (Smax) values of apigenin was 125 μM in RPMI medium and EC_50_ value was 72 μM in Jurkat cell treated with apigenin for 24 h [105]. It suppressed the JAK2/STAT3 pathway of gastric cancer cells in vitro and inhibited tumor growth in a xenograft model of gastric cancer in vivo [31]. Effects of apigenin on JAK2/STAT3 pathway and cytotoxicity in cancer cells were mediated by its docking to 26S proteasome in cancer cells [106]. Consequently, it potentiates the inhibitory effect of IFN-α, a potent antiproliferative factor, on cancer cell viability by activating JAK2/STAT3 signaling pathway through inhibition of 26S proteasome-mediated IFNAR1 degradation. Apigenin also induced p21^Cip1^ and NAG-1 through p53 activation in HCT116 colon cancer cells and its treatment reduced polyp formation via phosphorylation of p53 in APC^Min/+^ mouse models [32]. In addition, apigenin increased the sensitivity of doxorubicin-resistant hepatocellular carcinoma cells to doxorubicin by elevating miR-101 expression and suppressing Nrf2 expression, as a possible chemosensitizer [33]. Based on these reports, apigenin would be effective to treat gastric cancer and colon cancer through the activation of JAK2/STAT3 pathway and p53/p21^cip1^ pathway.

##### 2.2.1.2. Luteolin

Luteolin, found in various vegetables and fruits, induces apoptosis and cell cycle arrest and inhibited cancer cell migration, invasion, and angiogenesis [107]. Smax values of luteolin in RPMI medium was 25 μM [105]. It downregulated the PI3K/AKT-mediated signaling pathway of melanoma and carcinomas in breast, lung, and liver [40,41,42,107]. It also inhibited Nrf2 signaling to suppress the growth of non-small cell lung cancers in xenograft mice [43]. Downregulation of PI3K/AKT by luteolin reduced the growth of A375 melanoma cells in a xenograft mouse model [42]. Luteolin also increased the levels of intracellular ROS, thereby activating the expression of ER stress-related proteins and mitochondrial dysfunction, resulting in apoptosis of human glioblastoma U251MG and U87MG cells [44]. In silico molecular modeling, luteolin binds to the estrogen receptor ligand binding domain, showing anticancer effects with 25 μM IC_50_ values in MCF-7 cells [108]. These reports suggest that luteolin would be effective as an anticancer agent in melanoma, hepatoma, breast cancer, and non-small cell lung cancer.

##### 2.2.1.3. Oroxylin A

Oroxylin A, found in the roots of *Scutellaria baicalensis*, regulates apoptosis, angiogenesis, chemoresistance, metastasis, and invasion [109]. It induced apoptosis through p53 stabilization by downregulating MDM2, thereby inhibiting MDM2-mediated p53 degradation in hepatoma HepG2 cells [45]. Apoptosis induced by oroxylin A was associated with the functioning of Bax including its mitochondrial translocation, oligomerization, and activation. Oroxylin A promoted the engagement of Bax and Bcl-2 in human cervical cancer HeLa and colon cancer CaCo-2 cells [46,47]. In CaCo-2 cells treated with oroxylin A, inhibition of uncoupling protein 2 (UCP2) was responsible for hindering Bcl-2 translocation to mitochondria [47]. Additionally, the anti-tumor effect of oroxylin A was observed in a HeLa-transplanted mouse model [46]. Moreover, oroxylin A also effectively blocked metastasis of murine melanoma B16-F10 cells in vivo by suppressing matrix metalloproteinase-2/9 expression and blocking the ERK1/2 signaling pathway [48]. It inhibited ERK activity and activated GSK-3β, thereby inhibiting migration and invasion through degradation of Snail and suppression of epithelial-mesenchymal transition in non-small-cell lung cancer cells [49]. Similarly, suppression of migration and invasion by treatment of oroxylin A was observed in breast cancer MCF-7 cells under hypoxia, which involved suppressing the Notch pathway [50]. Chemoresistance is a hurdle in cancer treatment. Of note, oroxylin A enhanced the sensitivity of chronic myelogenous leukemia cells to imatinib and adriamycin by downregulating CXCL12/CXCR7, CXCR4, p-ERK, and the PI3K/AKT/NF-κB pathway in in vitro and in vivo mouse models of myelogenous leukemia K562 cells [51,52]. In addition, it improved the sensitivity of imatinib-resistant K562 cells by upregulating Stat3 and p-glycoprotein in vitro and in vivo in a K562 xenograft mouse model [53]. These anti-proliferation, pro-apoptotic, anti-metastasis, and chemosensitization effects of oroxylin A make it worthy of further study.

##### 2.2.1.4. Wogonin

Wogonin reduced colorectal cancer tumorigenesis through the nuclear translocation and stabilization of p53 [54,55]. It induced apoptosis by downregulating ERK/p38 MAPKs signaling or inhibiting PI3K-AKT signaling in breast cancer MCF-7 cells. It also increased the sensitivity of breast cancer cells to doxorubicin by inhibiting IGF-1R/AKT signaling [56]. By inhibiting Metalloproteinase (MMP)-9, wogonin suppressed the migration and invasion of hepatocellular, breast carcinoma, and osteosarcoma cells in vitro [58,59,60]. Additionally, wogonin exhibited anti-angiogenic activity by degrading HIF-1α protein and reducing the secretion of angiogenic growth factor VEGF in MCF7 cells and an in vivo xenograft-induced angiogenesis model [61]. Furthermore, in a non-small cell lung cancer xenograft mouse model, co-treatment of wogonin and TRAIL inhibited cFLIPL and IAP proteins [62]. Similar to its effect on solid tumors, wogonin was effective against leukemia. It exhibited apoptotic effects by blocking PI3K-AKT signaling or downregulating Bcl-2 and concomitantly reducing telomerase activity in HL-60 cells [57,103]. It inhibited Nrf2-dependent MRP1 expression in adriamycin-resistant human myelogenous leukemia K562/A02 cells [63]. These reports show that wogonin is a potent anticancer agent effective against solid tumors and leukemia.

#### 2.2.2. Flavonols with Anti-Cancer Effects

##### 2.2.2.1. Kaempferol

Kaempferol acts on EGFR/MAPK/AKT pathways for growth inhibition in human cervical cancer [64] and pancreatic cancer [66], showing an anticancer effect. Moreover, kaempferol inhibited the growth and metastasis in vitro and in vivo by inhibiting the expression of Bax, caspases, Fas, and cleaved-PARP and the activities of AKT, TIMP2, and MMP2 [65], which may be related with its docking to caspase according to recent crystal structure analysis [110]. Kaempferol also inhibits cancer cell growth by antagonizing estrogen-related receptor alpha and gamma activities [67], which would be related with its targeting estrogen receptor alpha in silico studies [111]. Its Smax value was 25 μM in RPMI medium and EC_50_ value was 163 μM in Jurkat cell [105]. Kaempferol induced autophagic cell death through the IRE1-JNK-CHOP pathway and by decreasing G9a in gastric cancer cells [68]. Thus, kaempferol induces cancer cell apoptosis, which may be related with its binding to caspase.

##### 2.2.2.2. Quercetin

Quercetin is a natural anticancer agent that is available at a lower price than conventional anticancer drugs [112]. The Smax of quercetin was 45 μM in medium and IC_50_ value was 354 μM in Jurkat cell [105]. In non-small-cell lung cancer A549 cells, quercetin was found to activate Bax/Bcl2-mediated apoptosis by abnormal microfilament disassembly and mitotic catastrophe [69]. Quercetin exhibits apoptotic effects in colorectal cancer HT-29 and breast cancer MCF-7 cells through its effects on COX-2 expression by suppressing AMPK, and regulating FasL via p51, p21^Cip1^, and GADD45 signaling [70,71]. Moreover, in breast cancer MCF-7 and MDA-MB-231 cells, quercetin treatment inhibited epithelial-mesenchymal transition determined by expression of markers such as vimentin, Snail, N-cadherin, Twist, Slug, MMP2, and MMP9, by suppressing EGFR/VEGFR2 signaling [72]. It also inhibits cancer stem cells as seen by its anti-tumorigenic effects on chemoresistant in pancreatic cancer stem-like cells through the suppression of β-catenin of the Wnt signaling pathway [73]. Its combination with gemcitabine reduces tumor growth and chemoresistance in pancreatic cancer stem-like cells.

##### 2.2.2.3. Myricetin

Its Smax was 700 μM and EC_50_ value was 349 μM in jurkat cell treated with myricetin for 24 h [105]. Myricetin inhibits cell proliferation, tumor metastasis, angiogenesis, and induces cell death by inhibiting cancer signaling pathways such as MAPK, AP-1/cyclin D1, JAK-dependent STAT3 in various cancer cell lines [113]. Moreover, it exhibits apoptotic effects by reducing VEGF expression and suppressing nucleoside diphosphate kinase in colon cancer HCT-15 cells [74]. As a docking site of myricetin, human flap endonuclease 1 was reported, based on structure based molecular docking study [75]. Its inhibition by myricetin increased the sensitivity of colon cancer [75]. It induces autophagy by inhibiting mTOR signaling in HepG2 cells [114]. Similarly, myricetin enhances the osteogenic differentiation of human periodontal ligament stem cells through the up-regulation of BMP-2/Smad and ERK/JNK/p38 MAPK pathway [76]. It inhibited cancer progression in human prostate cancer PC3 cells by inhibiting PIM1 and disrupting the PIM1/CXCR4 interaction [77]. It reduced intestinal tumorigenesis via the GSK3β and Wnt/β-catenin pathways in APC^Min/+^ mouse models [78].

#### 2.2.3. Flavanones with Anti-Cancer Effects

##### 2.2.3.1. Hesperidin

Hesperidin induces ROS-mediated apoptosis and cell cycle arrest in gall bladder carcinoma cells [79]. It was reported that hesperidin binds with PLK1 by computational analysis [115]. Thus, it would be possible the cell cycle arrest by treatment of hesperidin could mediate the inhibitory activity against PLK1. Moreover, it suppresses migration and invasion of non-small-cell lung cancer cells by inhibiting the SDF-1/CXCR-4 pathway [81]. Hesperidin exhibits anticancer effects by inhibiting the PI3K/AKT pathway in liver cancer cells [82]. In addition, hesperidin exhibits apoptotic effects through the Fas-initiated FADD/caspase-8 pathway in human lung cancer H522 cells [80]. In rat liver tissues, hesperidin inhibited carcinogen-induced hepatic carcinogenesis by activating the Nrf2/ARE/HO-1 pathway [83].

##### 2.2.3.2. Naringin

Naringin, obtained from tomatoes, grapefruits, and citrus fruits, exerts chemopreventive and anticancer effects in various models of oral, breast, colon, liver, lung, and ovarian cancer [116,117]. It suppresses cell growth through inhibition of NEU3 in HeLa and A549 cells and inhibition of the PI3K/AKT/mTOR pathway [84,85]. In addition, the suppression of cancer cell growth by naringin is mediated by the downregulation of HER2, MAPK, and Smad3/Smad7 signaling pathways [86,87,88]. Moreover, naringin exerts an anti-metastatic effect by suppressing cell invasion and migration through the inhibition of Zeb1 and MMP2/9 by acting on ERK-P38-JNK signaling [89,90,91]. Furthermore, in human endothelial cells, it exerts anti-angiogenic effects through the downregulation of ERRα/VEGF/KDR signaling and has apoptotic effects in human colon cancer by acting on ATF3 [92,93].

## 3. Flavonoids with PLK1 Inhibitory Effects

### 3.1. 7-O-Methylwogonin

7-*O*-methylwogonin, extracted from the roots of *Scutellaria baicalensis,* showed PLK1 inhibitory activity [22]. Traditionally *S. baicalensis* have been widely used as an anti-inflammatory and anticancer agent, owing to its inhibitory effects on the prostaglandin E production and cyclooxygenase activity [118,119,120]. The study revealed that 7-*O*-methylwogonin exhibited relatively selective inhibitory effects against PLK1 with an IC_50_ of 10.2 μM, based on kinase profiling using FRET and ADP-Glo kinase assays [22]. Kinase profiling showed that it has 70–100 fold greater selectivity for PLK1 over the majority of the kinases tested including PLK2, PLK3, Aurora A, Aurora B, CK1, CK2, DNA-PK, VRK2, EGFR, HER2, HER4, IGF1R, InsR, KDR, PDGFR, AKT1, PDK1, PKA, PKC, ROCK1, and RSK2 [22]. In this study, 7-*O*-methylwogonin suppressed the hepatoma Hep3B cell proliferation and induced mitotic delay, determined by immunostaining [22], which may be the result of its PLK1 inhibitory effects and modulation of mitotic cell division. However, the anticancer effects of 7-*O*-methylwogonin have to be investigated in detail.

### 3.2. Baicalein

Baicalein, isolated from the roots of *S. baicalensis* [121], significantly inhibited tumor volume and tumor weight in vivo [122]. It regulates cell proliferation [34], MAPK kinase signaling [35], PI3K/AKT pathway [39], and cancer metastasis [36]. Cotreatment of baicalein and silymarin arrests the cell cycle in the G1/S phase in hepatocellular carcinoma HepG2 cells by increasing p53, p21^Cip1^ and p27^Kip1^ expression [34]. Downregulation of MAPK, ERK, and p38 signaling by baicalein treatment in colon and breast cancer cells has also been reported [35,37,38]. Baicalein suppressed PLK1 activity with the IC_50_ value of 9 μM [22]. It showed low selectivity to PLK1 since it inhibited PLK3 activity with an IC_50_ of 40.3 μM, which was approximately 4.5-fold greater than its IC_50_ against PLK1, suggested that baicalein had relatively low selectivity among the polo-like kinases [22]. Treatment of baicalein induces cell cycle arrest at different phases, which depends on the cell type. Prostate cancer cells, osteosarcoma cells, hepatoma cells, and bladder cancer cells are arrested at the G1, G1/S, S, G2/M phases, respectively, by baicalein [34,123,124,125]. Thus, the anticancer effects of baicalein vary with the tissue or cell type.

### 3.3. Dihydrobaicalein

As one of the flavones isolated from *S. baicalensis*, dihydrobaicalein inhibits PLK1 kinase activity with an IC_50_ value of 6.3 μM [22]. Dihydrobaicalein is relatively more potent against PLK1 kinase activity than 7-*O*-methylwogonin, baicalein, and viscidulin II, isolated from the roots of *S. baicalensis.* However, dihydrobaicalein shows relatively low kinase selectivity because it inhibited vaccinia-related kinase 2 (VRK2) with an IC_50_ of 58.8 μM, which was just 9-fold higher than its IC_50_ against PLK1 [22]. While dihydrobaicalein has PLK1 inhibitory effects, its anticancer effects have not been investigated yet.

### 3.4. Viscidulin II

Viscidulin II inhibits the kinase activity of PLK1 with an IC_50_ value of 9.6 μM [22]. It also inhibited casein kinase 1γ1 and PLK2 with IC_50_ values of 114 and 125 μM, respectively, which were only 11- and 13-fold higher than the IC_50_ value for PLK1, respectively [22]. Thus, viscidulin II showed relatively low kinase selectivity. The anticancer effects of viscidulin II have not been studied yet.

### 3.5. Genistein

#### 3.5.1. Genistein as A Receptor Tyrosine Kinase (RTK) Inhibitor

Genistein, an isoflavonoid, regulates cell cycle progression, apoptosis, and metastasis. In 1987, genistein was identified as a specific inhibitor of receptor tyrosine kinase (RTK) including EGFR [94]. EGFR, a cell surface receptor having a TK domain, has been focused as a molecular target of cancer treatment, because its abnormal activation and overexpression have been observed in several malignant tumors [126]. Understanding the EGFR-mediated pathophysiology in cancer has led to the development of anti-EGFR agents including small-molecule inhibitors and monoclonal antibodies [127,128]. Genistein was discovered as an effective inhibitor against RTK including EGFR (Figure 2). EGFR kinase activity was inhibited by genistein using a radioisotope-based in vitro kinase assay with histone H2B substrate with an IC_50_ value of 22 μM [94]. Genistein prevents the autophosphorylation of EGFR in vitro with an IC_50_ of 2.6 μM [94]. Kinetic analysis showed that it is a noncompetitive inhibitor with histone H2B. In addition, cell culture experiments with epidermoid carcinoma A431 and Jurkat T-leukemia cells showed that genistein reduces the phosphorylation levels of RTK [94,96]. Based on these reports, genistein is considered to inhibit RTK.

#### 3.5.2. Genistein as A Direct PLK1 Kinase Inhibitor

Despite previous studies, its effects on cell cycle arrest are not fully explained. Some reports support that genistein does not inhibit EGFR phosphorylation in breast cancer [95] or prostate cancer cells following stimulation with EGF [129]. Second, genistein induces G2/M phase arrest instead of G1 arrest in most of the cancer cells reported [97,98,130]. Genistein induces mitotic catastrophe through mitotic arrest in several cancer cells, including carcinomas of the stomach, liver, lung, ovary, and breast [97,98,99,130,131,132,133,134,135]. Genistein induced cytokinesis failure by RhoA delocalization [133] and downregulation of KIF20A [98]. Treatment of its derivative ITB-301 disturbed microtubule polymerization and mitotic arrest [131]. The mitotic abnormality as a consequence of genistein treatment in cancer cells cannot be fully understood based only on its EGFR inhibitory effects. There is a possibility that genistein regulates the mitotic regulatory factors.

Recently, genistein was found to directly inhibit PLK1, a master mitotic kinase, based on an in vitro kinase assay [23] (Figure 2). The IC_50_ value of genistein against PLK1 activity was 7.9 μM, while that against EGFR activity was 64 μM [23]. In addition, the IC_50_ values of genistein against other TKs, such as erbB2, erbB4, IGF1 receptor, insulin receptor, kinase insert domain receptor, PDGFR, and PDGFR were over 4000 μM [23]. Genistein displayed 500-fold greater selectivity for PLK1 than other TKs tested, except for EGFR [23]. This study revealed that genistein selectively inhibits PLK1 activity, which is consistent with its effects on mitotic arrest and mitotic catastrophe [23], since genistein-treated cells were concentrated in the G2/M phase and sub-G1 fraction determined by FACS analysis. Immunochemistry showed that the cell population positive for p-histone H3 and MMP2, mitotic marker proteins, was higher in genistein-treated cells than in control, suggesting that genistein blocks mitosis as a PLK1 inhibitor.

#### 3.5.3. Genistein as A Suppressor of PLK1 Expression

Some studies showed that genistein suppresses the transcriptional expression of PLK1 indirectly in prostate cancer and neuroblastoma [100,134]. Genistein suppressed the expression of PLK1, which induced apoptosis and checkpoint activation. However, these effects seem to be indirect because the transcriptional factors involved remained elusive [100,134]. The suppression of PLK1 expression by genistein possibly resulted in apoptotic cell death and suppression of proliferation because genistein increased the expression of p53 and a CDK inhibitor p21^Cip1^, as well as Bax, an effector for apoptosis [100,134]. Consequently, the cell viability is reduced in genistein-treated cancer cells [100,134]. Thus, it is plausible that genistein reduces the activity and expression of PLK1 [23,100,134], which induces apoptosis because PLK1 depletion induces apoptosis [14,15] (Figure 2). Taken together, the transcriptional regulation of PLK1 by genistein is possibly an indirect effect.

#### 3.5.4. Genistein as A Modulator of Hormone Receptor

Structurally, genistein is similar to 17β-estradiol and hence, displays binding affinities to the ER, although its affinities are different for ERα and β [136,137]. The binding efficiency of genistein for ERα was 4% and that for ERβ was 87%, compared with estradiol [137]. Due to its structural similarity to 17β-estradiol and binding efficiency to ER, genistein has been studied as a natural ER antagonist or agonist. Genistein inhibits the binding of estrogens to ER, affecting estrogen metabolism and preventing estrogen-related cancer such as breast cancer. The anticancer effects of genistein against breast cancer have been studied in in vitro, in vivo, and through clinical experiments. Depending on the expression of ERα and/or β and the doses of genistein, the anticancer effects observed varies [101,138,139,140]. When breast cancer expresses ERβ, treatment with estrogen and genistein reduces cancer cell proliferation; however, relatively high concentrations of these compounds promote the proliferation of breast cancer cells expressing ERα [101,138]. Clinical trials with genistein as a supplementary agent in breast cancer patients showed different outcomes that depended on the patients’ menopausal status [139,141], suggesting that if genistein or soybean products are consumed too soon before or just around menopause, breast cancer cell proliferation may be accelerated. However, genistein or estrogen supplementation in estrogen-deprived conditions could be beneficial a decade after menopause [139]. Like estrogen, genistein may be useful or dangerous, depending on the estrogen status of breast cancer patients.

In addition, genistein has been studied for the treatment of prostate cancer. Androgens and androgen receptor (AR) function in the development and progression of prostate cancer [142]. Genistein transcriptionally downregulates AR, which results in the inhibition of prostate-specific antigen (PSA), a target of AR [102,143,144]. The effects of genistein on AR are dependent on androgen concentration and tissue specificity [102]. Like estrogen in breast cancer patients, genistein functions as an androgen antagonist in the prostate gland, testis, and brain in non-castrated males [102]. However, with a lack of circulating endogenous androgens in castrated males, genistein may function as an androgen agonist in the prostate gland and the brain [102]. A partial explanation is that selective AR modulators may stimulate the transcriptional regulation of distinct gene subsets in a tissue-specific manner. Thus, genistein could affect AR-mediated gene expression in a tissue-specific manner. While genistein is widely studied for cancer prevention, it should be also considered as a tissue-specific hormone modulator.

#### 3.5.5. Genistein in Clinical Trials for Chemoprevention and Cancer Treatment

By acting on multiple targets including PLK1, EGFR, and ER, genistein is a promising compound for cancer prevention and treatment. Its safety and kinetics have been demonstrated in a randomized double-blind phase 2 clinical trial for the treatment of prostate cancer [145] Fifty-four study subjects were treated with genistein (30 mg, daily, *n* = 23) and placebo (*n* = 24) for 3 weeks to 6 weeks prior to prostatectomy. The concentrations of total genistein in the plasma were on average 100-fold higher in the treatment arm than in the placebo arm (*P* < 0.001), and the adverse events in the treatment arm were few and mild. The efficacy was evident through reduced levels of PSA, and the normal and tumor tissue were comparable in the genistein-treated subjects. Currently, genistein is being tested in a phase II clinical study against adenocarcinoma of prostate cancer (NCT01126879; NCT02766478).

#### 3.5.6. New Contrivance for Overcoming the Hurdles of Genistein

While genistein shows significant biological anticancer effects, it still has several limitations such as low bioavailability and low solubility. To overcome these hurdles for its therapeutic usage, new contrivance has been tried. Nanocarrier-based delivery systems and synthesis of genistein analogues have been investigated [104,146,147]. When genistein is carried by nano-polymeric materials, its solubility and bioavailability were improved, which affects to the organ distribution and their therapeutic efficacy [146]. In addition, the synthesis of genistein analogues such as DFOG (7-difluoromethoxyl-5,4′-di-n-octylgenistein) and phenoxodiol exhibits potent anticancer activity compared with genistein [104,147]. These recent studies suggest that genistein is still a valuable agent for anticancer therapy.

## 4. Conclusions

Flavonoids including genistein are abundant in natural foods and are used in herbal medicine owing to their diverse effects including anticancer effects. They target critical oncogenic signaling pathways and kinases such as PI3K/ATK, EGFR, MAPKs, and PLK1. However, further basic and clinical studies are required to validate the use of these flavonoids in cancer treatment or chemoprevention. Genistein of all flavonoids is worth exploring further.

## Figures and Tables

**Figure 1 ijms-20-02420-f001:**
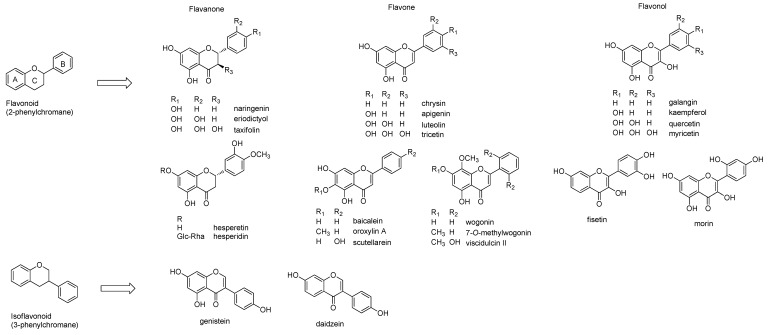
Structures of basic flavonoids and selected structures. Two main classes of flavonoids, Flavonoid (flavones, flavanones, and flavonols), and Isoflavonoid, are presented and additionally, modified flavonoids are frequently found in dietary and medicinal plants are described.

**Figure 2 ijms-20-02420-f002:**
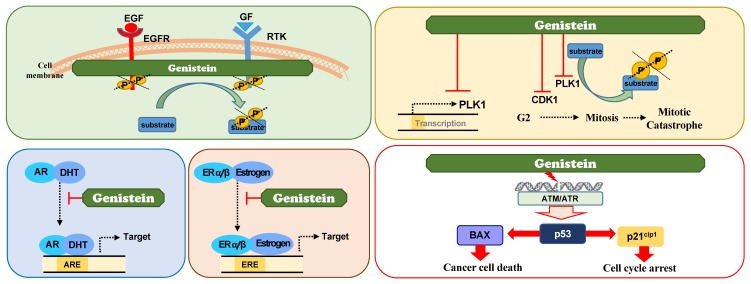
Action mechanisms of genistein to exert anticancer effects. Genistein exhibits anticancer effects by inhibiting the activities of receptor tyrosine kinase including EGFR and serine/threonine cell cycle regulatory kinases such as PLK1 and CDK1. Genistein also suppresses the transcriptional expression of PLK1. It is also considered as a tissue-specific hormone modulator.

**Table 1 ijms-20-02420-t001:** Molecular targets of flavonoids to exert anticancer effects.

Flavonoid (Class)	Molecular targets	Model	Ref
Apigenin (Flavones)	JAK2/STAT3	SGC-7901 (stomach), xenograft mouse model	[31]
	p53/p21^Cip1^	HCT-116 (colon), APC^Min/+^ mouse model	[32]
	Nrf2	BEL7402/ADM (liver)	[33]
Baicalein (Flavones)	Rb/E2F/cyclin-CDK4/p53	HepG2 (liver)	[34]
	MAPKs	HCT116, SW480 (colon), MG-63 (osteosarcoma) MDA-MB-231, MCF-7 (breast), xenograft model	[35,36,37,38]
	PI3K/AKT	HeLa, SiHa (cervix), MG-63 (osteosarcoma)	[38,39]
Luteolin (Flavones)	AKT	A375 (skin), A549 (lung), SK-Hep-1 (liver), xenograft mouse model	[40,41,42]
	Nrf2	A549 (lung), xenograft mouse model	[43]
	ROS/ER stress	U251MG, U87MG (glioma), xenograft mouse model	[44]
Oroxylin A (Flavones)	p53	HepG2 (liver)	[45]
	Bcl-2	HCT-116 (colon), HeLa (cervix), xenograft model	[46,47]
	ERK/MAPK	MDA-MB-231 (breast), A549 (lung)	[48,49]
	Notch pathway	MCF-7 (breast)	[50]
	CXCL12	K562, KU812, xenograft mouse model	[51,52]
	STAT3	K562, xenograft mouse model	[53]
Wogonin (Flavones)	p53	HCT-116, xenograft mouse model	[54,55]
	PI3K-AKT	MCF-7 (breast), HL-60 (leukemia)	[56,57]
	MMP-9	MHCC97L, PLC/PRF/5, CD133+CAL72	[58,59,60]
	HIF-1α	MCF-7, MDA-MB-231 (breast), xenograft model	[61]
	cFLIPL and IAP	A549 (lung), xenograft mouse model	[62]
	Nrf2/ARE	K562/A02 (leukemia)	[63]
Kaempferol (Flavonols)	PI3K/AKT	HeLa (cervix), HCCC9810, QBC939 (liver), xenograft mouse model	[64,65]
	EGFR/ERK	Miapaca-2, Panc-1, SNU-213 (pancreas)	[66]
	ERRα/γ	HeLa (cervix), HepG2 (liver), A549 (lung)	[67]
	IRE1-JNK-CHOP	AGS, SNU-638 (stomach)	[68]
Quercetin (Flavonols)	BCL2/BAX	A549 (lung)	[69]
	AMPK/COX-2	MCF-7 (breast), HT-29 (colon)	[70]
	p53	MDA-MB-231 (breast)	[71]
	EGFR/VEGFR2	MCF-7, MDA-MB-231 (breast)	[72]
	β-catenin	Pancreatic cancer stem-like cells	[73]
Myricetin (Flavonols)	NDPK	HCT-15 (colon)	[74]
	mTOR/AKT	HepG2 (liver)	[75]
	BMP-2/Smad, MAPKs	Human periodontal ligament stem cells	[76]
	PIM1/CXCR4	PC3, DU145 (prostate), xenograft mouse model	[77]
	Wnt/β-catenin	APC^Min/+^ mouse model	[78]
Hesperidin (Flavanones)	p53, Bax, caspases-3	H522 (lung), gall bladder carcinoma cell	[79,80]
	SDF-1/CXCR-4	A549 (lung)	[81]
	PI3K/AKT	Rat (liver)	[82]
	Nrf2/ARE/HO-1	Rat (liver)	[83]
Naringin (Flavanones)	AKT/mTOR	H69AR (lung)	[84]
	EGFR, NEU3	A549 (lung)	[85]
	HER2	SK-BR-3, MDA-MB-231 (breast)	[86]
	Smad3/Smad7	Mouse melanoma	[87]
	MAPKs	JAR, JEG-3 (placenta) U87, U373, U251 (glioma)	[88,89,90]
	Zeb1	MG63, U2OS (osteosarcoma)	[91]
	ERRα/VEGF/KDR	HUVECs (endothelium)	[92]
	ATF3	HCT116, SW480 (colon)	[93]
Genistein (Isoflavonones)	EGFR	A431 (skin), MCF-7, BT20, ZR-75-1 (breast)	[94,95]
	CDK	Jurkat T (leukemia), MCF-7, MDA-MB-231 (breast)	[96,97]
	KIF20A	SGC-79019 (stomach)	[98]
	PLK1	MCF-7, BT20 (breast), H1299 (lung), HeLa (cervix) HepG2 (hepatoma), LNCaP, PC-3 (prostate)	[23,99,100]
	ER α/β	MCF-7 (breast)	[101]
	AR	Mouse tissues	[102]

**Table 2 ijms-20-02420-t002:** Common signaling pathways triggered by flavonoids for anticancer effects.

Signaling Pathway	Flavonoids	Ref
MAPK pathway	Baicalein Oroxylin A Myricetin Naringin	[36,37,38] [49] [76] [88,89,90]
PI3K/AKT pathway	Baicalein Luteolin Wogonin Kaempferol Hesperidin Naringin	[39] [40] [56,57] [64,65] [82] [84]
p53 pathway	Apigenin Baicalein Oroxylin A Wogonin Kaempferol Quercetin Hesperidin Genistein	[32] [34] [45] [54,55] [64] [71] [80] [99]
Apoptosis	Oroxylin A Wogonin Kaempferol Quercetin Myricetin Hesperidin	[46,47] [103,104] [64,65] [69] [74] [79,80]

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
