# Peer review of "Molecular Targets of Genistein and Its Related Flavonoids to Exert Anticancer Effects"

_ijms, 2019, doi:10.3390/ijms20102420_

Round 1

Reviewer 1 Report

The authors paid special attention to anti-cancer effects of flavonoids in various types of cancers. In addition, they showed that relationships between such anti-cancer effects and flavonoids target signaling pathways at molecular levels. Finally, treatment strategies used genistein, which is multi-functional flavonoid as receptor tyrosine kinase inhibitor, polo-like kinase 1, and modulator of hormone receptor, are introduced.

I think this review has important information to discuss pathological significance of genistein and its related flavonoids and treatment strategies used them in various types of cancers. In fact, I have no major question. However, I would like to know your opinion about followed suggestion.

(Minor)

In 3.5.5. Genistein in clinical trials for chemoprevention and cancer treatment section, you introduced clinical trails for prostate cancer. We recommend adding the information on new contrivance to improve the anti-cancer effects of genistein-based therapy (For examples; biocompatible nanocarrier-based genistein delivery systems and Genistein Analogues) from previous reports.

Author Response

Response to comments of reviewer 1

Original Comment

In 3.5.5. Genistein in clinical trials for chemoprevention and cancer treatment section, you introduced clinical trials for prostate cancer. We recommend adding the information on new contrivance to improve the anti-cancer effects of genistein-based therapy (For examples; biocompatible nanocarrier-based genistein delivery systems and Genistein Analogues) from previous reports.

Authors’ response

We appreciate reviewer’s comment. According to the reviewer’s comment, we add the information on new contrivance to improve the anti-cancer effects of genistein-based therapy such as nanocarrier-based delivery and synthesis of genistein analogues at the end of text in page 10, as shown below.

3.5.6. New contrivance for overcoming the hurdles of genistein

Although genistein shows significant biological anticancer effects, it still has several limitations such as low bioavailability and low solubility. To overcome these hurdles for its therapeutic usage, new contrivance has been tried. Nanocarrier-based delivery systems and synthesis of genistein analogues have been investigated [145-147]. When genistein is carried by nano-polymeric materials, its solubility and bioavailability were improved, which affects to the organ distribution and their therapeutic efficacy [145]. In addition, the synthesis of genistein analogues such as DFOG (7-difluoromethoxyl-5,4’-di-n-octylgenistein) and phenoxodiol exhibits potent anticancer activity compared with genistein [146-147]. These recent studies suggest that genistein is still a valuable agent for anticancer therapy.

Reviewer 2 Report

In the review article titled "Genistein and its related flavonoids target PLK1 and associated signaling pathways to exert anticancer effects" authors summarized anti-cancer activity of flavonoids and isoflavonoids and their cellular targets. The authors specifically focused on a isoflavonoid, genistein.

Authors summarized different flavonoid framework and picked few examples from each group and discussed their cellular targets.

Major comments:

The review was written only to present published facts but not to analyze any of the information.

It was not clear from the introduction and the body of the manuscript why genistein was particularly emphasized, especially because the IC50 values are in the 20-60 micromolar range (not significantly potent).

Authors discussed several subclass of flavonoids but they need to summarize if there is a specific trend from each subgroup towards a cellular target. For example does all flavones show similar cellular targets?

Authors should include any molecular docking/modeling/computational data that can compare different subclass.

Authors should discuss solubility and cytotoxicity of these molecules.

Minor comments:

Both tables need to be formatted.

Author Response

Response to comments of reviewer 2

Original Comment 1.

 In the review article titled "Genistein and its related flavonoids target PLK1 and associated signaling pathways to exert anticancer effects" authors summarized anti-cancer activity of flavonoids and isoflavonoids and their cellular targets. The authors specifically focused on an isoflavonoid, genistein. Authors summarized different flavonoid framework and picked few examples from each group and discussed their cellular targets. The review was written only to present published facts but not to analyze any of the information.

Authors’ response

         We appreciate reviewer’s comment. According to reviewer’s request, we have tried to analyze the information such as common targets and signaling pathways triggered by flavonoids as shown in Table 1 and additional Table 2. Moreover, the title has been rewritten as “Molecular targets of genistein and its related flavonoids to exert anticancer effects” to make concise and clear.

Original Comment 2.

 It was not clear from the introduction and the body of the manuscript why genistein was particularly emphasized, especially because the IC50 values are in the 20-60 micromolar range (not significantly potent).

Authors’ response

We appreciate reviewer’s comment. Among several flavonoids having PLK1 inhibitory effects, genistein was more selective and effective to PLK1 with IC50 7.9 mM against PLK1 kinase activity based on an in vitro kinase assay [section 3.5.2]. In addition, genistein has been broadly studied for the prevention or treatment of cancer including clinical studies. Because of these reasons we focused on genistein than other flavonoids.

To make clear the reason why we focused on the effect of genistein, the end of introduction was rewritten in page 2, lines 50-51, as below.

Flavonoids from natural products have PLK1 inhibitory effects. Genistein, baicalein, 7-0-methylwogonin, dihydrobaicalein, and viscidulin II have PLK1 inhibitory activity [22, 23]. Among them, genistein is more selective and effective to PLK1 compared with other flavonoids. In the latter part of this review article, we discuss the PLK1 inhibitory activities of flavonoids especially those of genistein. In addition, the anticancer effects of genistein through its effects on diverse targets are reviewed, since genistein has been broadly studied for chemoprevention and chemotherapy.

Original Comment 3.

 Authors discussed several subclass of flavonoids but they need to summarize if there is a specific trend from each subgroup towards a cellular target. For example does all flavones show similar cellular targets?

Authors’ response

We appreciate reviewer’s point. The similar cellular targets such as MAPK and PI3K pathways were collected in Table 2, although the relationship between their structures and targets was not unrevealed.

Original Comment 4.

Authors should include any molecular docking/modeling/computational data that can compare different subclass.

Authors’ response

According to reviewer’s point, the molecular docking, modeling, or computational data if it is reported, the data was added in each section of flavonoids.

Original Comment 5.

 Authors should discuss solubility and cytotoxicity of these molecules.

Authors’ response

The solubility and cytotoxicity of the flavonoids were added in each section of flavonoids.

Original Comment 6.

 Table need to be formatted.

Authors’ response

Table 1 was reformatted and Table 2 was newly made for the analysis of common pathways triggered by flavonoids.

Round 2

Reviewer 2 Report

Authors have addressed most of the concerns except computational studies. I believe currently the content is appropriate for the journal for publication.